# How doctors build community and socialize into a clinical department through morning reports. A positioning theory study

**Jane Ege Møller**[1]*, **Mads Skipper**[2], **Lone Sunde**[1], **Anita Sørensen**[3], **Thomas Balslev**[4], **Pernille Andreassen**[5], **Bente Malling**[1]

**1** Department of Clinical Medicine, Aarhus University, Aarhus, Denmark, **2** Region North, Viborg, Denmark, **3** Bispebjerg & Frederiksberg Hospital, Capital Region of Denmark, Copenhagen, Denmark, **4** Department of Clinical Medicine, Regional Hospital, Viborg, Aarhus University, Aarhus, Denmark, **5** The Danish National Center for Obesity, Central Denmark Region, Aarhus, Denmark

* jane@clin.au.dk

**Data Availability Statement:** The data set contains potentially identifying patient information and therefore we are not able to share it. This is in

## Abstract

### Phenomenon

The morning report is one of the longest surviving hospital practices. Most studies of the morning report focus on the effectiveness of formal medical training, while focus on social and communicative aspects is rarer. This study explores the social interactions and communication in morning reports, examining the ways in which they contribute to the construction of professional identity and socialization into the community of the clinical department.

### Approach

We used a qualitative explorative design with video observations of morning reports. Our data consisted of 43 video-recorded observations (in all, 15.5 hours) from four different hospital departments in Denmark. These were analyzed using the theoretical framework of positioning theory.

### Findings

A key finding was that each department followed its own individual structure. This order was not articulated as such but played out implictly. Two alternative storylines unfolded in the elements of the morning report: 1) being equal members of the specialty and department, and 2) preserving the hierarchical community and its inherent positions.

### Insights

The morning report can be seen as playing an important role in community making. It unfolds as a "dance" of repeated elements in a complex collegial space. Within this complexity, the morning report is a space for positioning oneself and others as a collegial "we", i.e., equal members of a department and specialty, at the same time as "having a place" in a hierarchal community. Thus, morning reports contribute to developing professional identity and socialization into the medical community.

accordance with the Danish Data Protection agency regulations. Contact information: Databeskyttelsesenheden, e-mail: dpo@au.dk, Aarhus University, Navitas, Inge Lehmannsgade 10, Building 3210, DK-8000 Aarhus C.

**Funding:** No grant number available Name funder: Central Denmark Region URL:https://www.rm.dk/sundhed/faginfo/uddannelse/sundhedsuddannelser The funders had no role in study design, data collection and analysis, decision to publish, or preparation of the manuscript.

**Competing interests:** The authors have declared that no competing interests exist.

## Introduction

The morning report is one of the longest surviving hospital practices and is still widespread. However, it has undergone changes over time [1,2]. While previously it was primarily used to monitor daily patient care and assessment of trainees, the morning report is now typically described as a more complex event that includes patient care-related activities, educational elements that are often case-based, patient handover, and work planning [2–5].

Studies show that many (especially junior) hospital doctors value the morning report as a cornerstone in medical education and see it as a space for learning through cases and patient handover [6–10]. In their study, Albert et al. found that internal medicine residents consistently value the opportunity provided by morning reports to discuss compelling cases, practice clinical reasoning and have group discussions in a supportive learning environment [8].

By contrast, others point out that morning reports are superficial and demand a lot of resources because of the lack of learner-centered frameworks [11]. This discrepancy in the way in which the morning report is valued is also observed in a review that concludes that the morning report has heterogeneous aims, forms and settings, making it difficult to measure learning outcomes [5]. Thus, although many see the morning report as an important educational activity, its learning value remains unclear. In their study, Heppe et al. investigated the format and content of internal medicine morning reports in the US and found that the morning report predominantly consists of case-based presentations [12]. However, they argue that learning outcomes could be improved by implementing principles from adult learning theory, e.g., co-learning [8].

Hill et al. conclude that the preservation of the morning report is due to its cultural value rather than its learning possibilities [11]. They describe the morning report as a cultural *rite de passage* to entering the medical profession, and thus as nothing but a cultural and social ritual. It has been shown how the collegial interaction and communication that take place in the morning report are part of the construction of a doctor's professional identity, where residents learn to replace their "life-world" voice with the "voice of medicine" [7]. More generally, medical education research has drawn attention to social and cultural aspects of medical education. Some studies have shown that senior physicians' social influence indirectly affects trainees' learning [13,14]. Social identity theory as an approach in medical education research also directs attention to how social processes influence medical education and professional identity formation [14–16]. Accordingly, a theoretically based framework has been proposed as a way of understanding professional identity at the micro-level, which involves behaving in accordance with a health profession's (tacit) cultural dimensions in order to be acknowledged as a member [17]. Other studies have provided micro-level analyses of the interactions involved in handover and simulation training [18,19]. Despite this, cultural and social aspects of the morning report are not explored in much detail, even though it is supposedly a central space for informal socialization [6]. In this study, we wish to explore if and how the morning report may play a role in such socialization, creating a sense of community and professional identity.

Most studies of the morning report are interview studies or surveys that focus on the effectiveness of formal training, especially case presentations [20–25], while a focus on social and cultural aspects is rarer, leaving these aspects of the morning report unexplored. Against this backdrop, this study explores 1) how social interactions and communication are displayed in morning reports, and 2) what that means for the formation of professional identity and socialization into the clinical department's community.

## Setting

In Denmark, which is the context of this study, the form of the morning report involves more activities compared to other countries. While the morning report in the North American

tradition, for example, primarily relates to internal medicine and is exclusively based on case-based educational sessions [8,16,26], the morning report in Denmark is a slightly broader event, best described as a morning meeting called 'morning conference' ('morgenkonference' in Danish). It is a mono-professional meeting held as the first thing in the morning for all doctors at most Danish hospital departments (not only internal medicine). All doctors who are at work are supposed to participate in the morning report. Besides case-based learning and short presentations, it includes elements such as handover and care planning. This broader scope of the morning report makes Denmark an interesting case to study as it displays a concentration of a variety of social interactions.

To better understand our study, it is relevant to know that the healthcare system in Denmark is primarily public, and general practitioners are gatekeepers for secondary and tertiary care. As a citizen, you need a referral from your general practitioner to be admitted to a hospital. The hospitals have limited capacity, and there are generally waiting lists. However, by law, various "packages of care" have been introduced. Thus, a patient with symptoms of serious or acute illness (for example cancer) can bypass the waiting list in order to be treated fast. Likewise, there are rules ensuring that as a patient, your symptoms will be considered within 30 days from referral. The hospitals are financed by one of five regions and therefore, the regional authorities prefer to keep treatment inside the region, since treatment in another region would entail extra payment. There can be an inbuilt conflict between doctors, who want the best treatment for their patients, and the management, who wants the most treatment for the finances they have at disposal from the government.

## Methods

The study used a qualitative design with an explorative focus. We used video-recorded observations of morning reports in different hospital departments.

### Material and participants

In order to capture characteristics across specialty thresholds and aim for the principle of maximal variation, departments representing a spectrum of specialties and hospitals were invited. Video recordings were made in the four departments where the head of department accepted the invitation: an emergency department, an internal medicine department and two different surgery departments [27]. Convenience sampling (time and geography) was used to choose the specific morning reports where the aim was to have at least 10 recordings from each department. The first author and two research assistants carried out the video recordings, and introduced themselves and the project briefly each morning unless the head of department did so (see Table 1 for an overview of the participating departments).

### Analysis

The analysis followed five steps (see Table 2 for a detailed description).

The use of an explorative and iterative analytical process made it evident that morning report elements were repeated in an almost scripted way (step one). To analyze this in depth, we separated the repeated actions and elements in a structured template. This made it possible to capture key repeated patterns of the discursive interactions in the observed morning reports and transform them into the template shown in Table 3 (step two). The template includes all elements detected in the morning reports; however, it does not represent a specific timeline or practice revealed by the data. Thus, the order in the template does not necessarily represent the chronology in which the morning reports were carried out. The template was then used to code the video recordings into various parts of the morning conference (steps three and four).

**Table 1. Overview of departments and data collection.**

|  | Consultants | Certified specialists | Registrars | Junior doctors | Total number of doctors | Number of video recordings (hours) | Number of participants * (average) |
|---|---|---|---|---|---|---|---|
| Emergency | 7 | 3 | 4 | 12 | 26 | 12 (5,7) | 8 |
| Surgery 1 | 20 | 12 | 15 | 5 | 52 | 13 (3,1) | 25 |
| Surgery 2 | 12 | 6 | 6 | 2 | 26 | 7 (2,3) | 15 |
| Internal medicine | 16 | 10 | 20 | 14 | 60 | 11 (4,4) | 30 |
| Total | 62 | 31 | 45 | 33 | 164 | 43 (15,5) | - |

*The distribution between consultants, specialists, trainees, and students follows the work plan; in all recordings, all levels, from students to consultants/specialists and in most cases the head of department, were represented.

To analyze the social interactions and communication unfolding *within* these elements, we used positioning theory as a theoretical strategy [29]. Positioning theory is an analytical tool for understanding intentional interactions in social episodes "under a local moral order, and the local system of rights and obligations" [30].

This analytical approach has three core concepts, known as the positioning triangle: 1) the position, 2) the action, and 3) the storyline (see Fig 1). A position is defined as a cluster of *rights* and *obligations* to perform certain actions [30]. Any social environment has a range of positions that people may adopt and in which they may try to locate themselves, or may be pushed into, or move away from [31]. Actions include both speech acts and other actions. By including speech acts, positioning theory explicitly draws on the notion of statements as performative from speech act theory [29,32]. However, the focus is broader and more dynamic, as it includes all intentional actions and their dynamic relation to social orders. Any significant social action (including speech) must be interpreted as a particular kind of action in relation to a specific social episode and its cultural norms. Thus, this practice organizes actions as 'actions of certain kinds'. Shaking hands for example, may signify 'a greeting', or 'goodbye' 'confirming an agreement', or 'congratulations'—the meaning of the action is determined by the social order. The storyline orders the episode. Social episodes do not develop randomly but follow already established patterns of movement, i.e. narrative conventions, i.e., *the storyline* [29]. Sometimes storylines are quite conventional such as 'Doctor—patient'. This encapsulates what

**Table 2. Overview over data analysis.**

| Step | Activity |
|---|---|
| 1 | All authors watched six randomly selected video-recorded morning reports, while sitting together to allow for discussion. During each video recording, each author wrote down individual reflections. At the end of each video, notes were shared, and preliminary themes and elements were identified. |
| 2 | Two authors (first and sixth) watched another six video recordings and made a template for analysis that included all themes and separated the repeated elements. |
| 3 | The author group met four times to watch and analyze additional video recordings using the template and adjust it along the way. The template was finalized. |
| 4 | Each author analyzed the rest of the video recordings (4–6 each) individually using the template. Then the author group met and shared notes with all authors. |
| 5 | In order to reach different levels of interpretation [28], theoretical analysis using the framework of positioning theory was made by the first author. In this phase, all elements in the template were analyzed in order to identify which storylines, positions and (speech) acts were observed in the material. In this process, the first author and a research assistant transcribed selected passages verbatim. Two main storylines were identified (see Results). These were then discussed in the group and related to the template in a process where some elements were merged. |

**Table 3. Template for data analysis.**

| Element | Observation | Reflection |
|---|---|---|
| Prelude | | |
| Opening | | |
| Patient handover/report | | |
| Asking advice/counselling | | |
| The role of the newcomer to the report | | |
| Responsibility for changing the topic | | |
| Physical placement | | |
| Announcements | | |
| Formal teaching | | |
| The working plan | | |
| Who is absent | | |
| Closing | | |
| The minutes after | | |
| Improvised use of humor | | |

can be expected by the people in the episode and which actions must be given meaning as actions and in what way.

Central to this framework is the idea that a person can take or be in two different positions at the same time, e.g., you can be both a 'patient' and 'company director'. Consequently, different storylines may be at work simultaneously in the same episode [29].

Positioning theory has previously proven to be a useful analytical lens in medical education and clinical contexts [33,34]. It is thus suitable to grasp the dynamics of social interactions in clinical workplaces, such as the morning report (Step five).

In Analysis Step 5, the results were condensed into clusters not necessarily following the template used in the initial analysis. Thus, some of the elements in the template were merged, while others were reformulated to create meaning and others were maintained. This synthesis resulted in seven headings describing the morning report practice revealed by the data combined with an interpretation of what this practice means for the development of professional identity and socialization into the medical community in the department.

## Ethics

All participants in the recorded morning reports received oral and written information about the project, and written consents were obtained. One doctor did not wish to participate in the project, and his statements were excluded. All transcripts were anonymized. All participants were anonymized using the following abbreviations: junior doctors were referred to as JD followed by a number referring to the video recording–e.g.: JD1 and JD5; medical specialists and consultants were referred to as SP followed by a number; and the Head of Department was referred to as HD followed by a number. The departments were anonymized using the following suffixes: internal medicine = IM; surgical department = S; and emergency department = EM–all followed by a number.

As patients only participated indirectly in the research project, we were exempted from obtaining written consent from them. The study was exempted from ethics approval according to the Act on Research Ethics Review of Health Research Projects. The local research committee was notified, and the Danish Data Protection agency approved the study (record number 2016-051-000001, no. 1782).

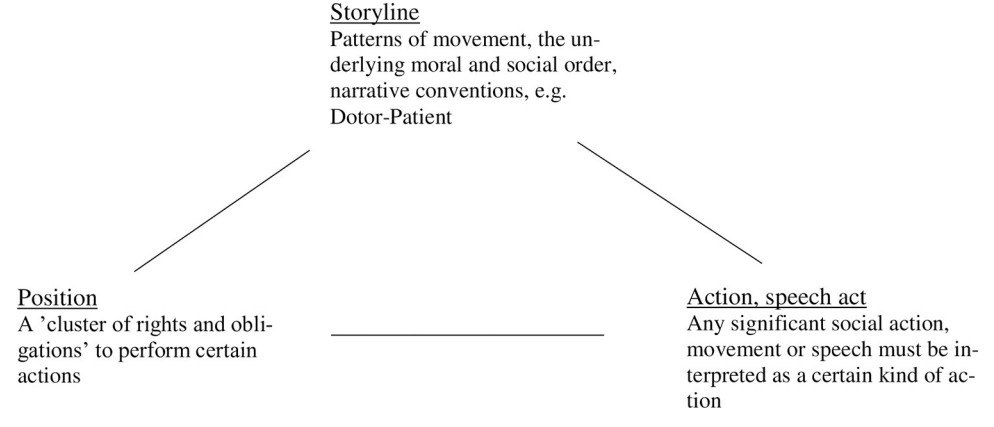

**Fig 1. The positioning triangle.**

## Results

In total, 43 morning reports were recorded (15.5 hours). Table 1 shows the number of recordings doctors employed at each of the four departments and the average number of participants in the recordings. The typical report duration was 15–20 minutes, varying from approximately 7 minutes to 45 minutes. The latter was unusual–when, for example, a department had allocated teaching time for a guest lecturer.

A key finding was that each department followed its own individual practice in all reports. Each department had its own order, despite variations between the departments, in the sequence in which the different elements unfolded; positioning, e.g., who conducted the progression of events (the chief physician or the on-call physician); and actions, e.g., the way in which openings and closings took place. This order was not articulated explicitly, but played out as an implicit or tacit structure.

Two different storylines became evident in the material and unfolded in the morning report: 1) preserving the hierarchical community and its inherent positions, and 2) being equal members of the specialty and department. These will be described in further detail below.

### Prelude and opening

The prelude was the short period before the opening of the morning report. In these minutes, informal chats were witnessed as doctors entered the room, where some put on coats, and others arranged their hair while finding their place and sitting down. Phrases like "Did you get some sleep last night?" or "Were you busy last night?" to the on-call physician were heard. Checking on how things went with a patient led to short dialogues such as "How is Mr. xx with xxx doing? Well, radiology was busy so we didn't get the scan". In addition, we observed doctors asking about or sharing aspects of their private life, such as "Do you always ride your bike to work?" Speech acts involving requests for help with matters such as trainees figuring out how to get competencies assessed were also seen. Such examples of informal collegial chat and small talk can be interpreted as contributing to the creation or maintenance of interpersonal relations and as a socialization process.

In addition, frustrating experiences that did not relate specifically to patient treatment were shared. This is evident in the following example:

| | This departing on-call doctor is telling how the night shift went to a small group of colleagues as they wait for the morning report to start. The departing on call doctor is a Danish junior doctor of south Asian origin. |
|---|---|
| JD1 | We discharged someone that refused to leave the department. The wife of the patient started off by saying to me: 'Are you, uh—are you educated in Denmark?' |
| Everyone | Laughing |
| JD1 | Get the message–oh, shut up. (as mock response to the wife) |
| Everyone | Laughing |
| JD1 | 'I almost said to her: 'Yes, I just came in yesterday on a ship from Uganda.' |
| SP1 | Laughing—You should have said that. |
| JD1 | 4 o'clock in the morning, man, I am just like 'Is that what you want to ask the doctor right now?' |
| JD2 | Are you sure you want to talk about this right now?' (ironic mock dialogue with the wife) |
| SP1 | Laughing—What time was it, was it late at night? |
| JD1 | Around 4 a.m. |
| JD2 | No, it's simply not okay. |
| SP1 | But then it's also a bit dark (humor). |
| Everyone | Laughing |
| JD2 | How rude. |
| SP1 | It's not OK. |
| JD1 | Yes—and she refused to leave the ward. The husband had to go home because she couldn't get her night's sleep. |
| JD2 | Just remember that next time. (ironically) |
| JD1 | Yes, I need to upgrade my Danish first. |
| Everyone | Laughing |

As observed, the act of informal, collegial sharing took place in this interaction. The resident's story seemed to unite the colleagues in mockingly distancing themselves from what they understood as the wife of the patient's racist and selfish utterances. The prelude, as part of the social episode, thus offered an opportunity for the speech acts of sharing other aspects of the night shift than those that were presented at the report itself, as well as small-talk and the exchange of experiences that could enable collegial support and contribute to creating interpersonal relations.

The sharpness of the demarcation between the prelude and opening varied from department to department. In some cases, the person conducting the morning report (who was either the on-call registrar, the head of department or another senior doctor) made the opening speech act for example, "Good morning, let's start", or as in the case of two of the departments a bell with a "ding" marked the beginning. In some departments, coming in late was a deviation from the norm. Others had more fluid transitions with doors opening and people entering, some putting on their coats in the conference room even after the morning report had started. The opening, nevertheless, marked a transition in the storyline—from informal prelude to formal interaction. As seen above, the speech acts in the prelude primarily abided by the norms of "How one is a colleague, e.g., sharing something about one's private sphere, helping out and expressing concern for fellow colleagues". The opening "ding" or "good morning" marked a shift in positions from an informal "room of equal colleagues" to a "community of hierarchy" with formalized discursive interaction.

## Physical setting and distribution of speech

Although none of the departments had explicit seats for the participating doctors and students, seating followed certain patterns. All departments had two circles of chairs, an inner circle at

the table and an outer "circle" with chairs against the wall. If there were medical students in a department, these and in some cases the youngest doctors would sit in the outer circle, and the more senior doctors would sit in the inner circle. In two departments, the head of department sat at the same place (at the end of the table closest to the door). In other departments, the head of department moved a couple of seats from day to day, but within a limited area.

It was observed that certain places in the room were also positions, as they related to speech rights and obligations. Apart from the on-call resident who had the right and duty to report, speech activity adhered to the following hierarchy: consultants spoke the most, and the more junior the doctor, the less they spoke. Because of the static seating pattern, speech was observed as mainly taking place in certain parts of the conference room. Seats in the outer circle seemed to entail that the person in that seat did not have the right to speak. The seating pattern thus seemed to constitute a physical positioning that reflected and reproduced the storyline of the hierarchical community.

## Outside and inside the department

The mentioning of people "inside" and "outside" the department created positions. More specifically, two different kinds of inside/outside relations were observed. One was *external collaborators* of various kinds, whether it was nursing staff, fellow physicians in other departments, general practice doctors, hospital management, cleaning staff, etc. Another was *the flow–in and out of the department* of people who either started or ended their training/working period, e.g., medical students and residents.

Speech acts concerning *external collaborators* mostly marked a "them and us". In some instances, this was demonstrated through distancing oneself from messages to be delivered, such as this doctor: "The cleaning staff asked me to say that we need to tidy the room more. There now–I've said it!" (IM). In another example, a third-year registrar (JD3) reminded all colleagues what to do when their emergency plan test would take place in a couple of days.

---

"Can I just use half a minute? You all received this (holds a piece of paper in his hand) in your email. It contains instructions for the test of our emergency plan next week." He then goes on to give a bit more detailed information about what the test involves. When he stops, all colleagues silently look at him. He then breaks the silence by saying: "It wasn't my decision. Don't shoot the messenger!" They all laugh. (JD3)

---

By making the "don't shoot the messenger" joke, JD3 seemingly positioned himself as not responsible for this exercise and the colleagues as acting unfairly. By laughing, the colleagues probably recognized that they appeared to have been positioning him as one of the outsiders (e.g. administrators) who introduce such exercises. Laughing at his joke thus may function as recognition of him as being a member of the group of equal colleagues.

Another positioning pattern related to colleagues in other places in the healthcare system. This was, for example, seen in joking mockingly about other departments not fulfilling their responsibility, for example "Don't you think that the pulmonary medicine physicians owe us cake by now?" (IM1). Such speech acts established a distinction between "us and them". A predominant feature of this pattern was that the outsiders were positioned as someone "not as good as", "not working as hard as", "not as knowledgeable as", or "performing as well as" us. A certain type of "we" was thus created through the positioning and ordered by the storyline of being a member of the group of doctors in this department.

At times, this "we" took the form of unity in the battle for local practices and treatments. This was observed in a sequence where a department discussed their struggle to fulfill a nationally decided maximum waiting time for a certain patient group. In the following example, a consultant (SP4) started with the comment "I have a service announcement" and then informed colleagues that they should inform their patients about a probable delay which would conflict with their legal right to treatment within four weeks.

| |
|---|
| One consultant (SP5) responded by sharing a story of one of his patients who did not accept a postponement but kept on pushing for an earlier appointment and got it. By doing so, the patient jumped the queue at the expense of another patient. |
| SP5: It's going to be at the expense of another patient. |
| SP6: No offense, but we have a five-week waiting list now – |
| SP7 What do we do if patients do not accept it? |
| SP3: Previously, we were told that we're not allowed to refer these patients to another hospital. |
| SP4: On Wednesday, we will have a report where we prioritize which ones [patients] we treat here and which we refer to other hospitals. We have been asked to do that. And we're doing this on Wednesday when Anna is back. We have been told to do that (. . .) |
| SP7: Not to be going on about it, but if we have a patient who will not accept [a later treatment] and who we are not allowed to refer, you're saying that we must find time for them, which means that we'll harm another. . . |
| SP7: Yes, it does. |
| SP5: Patient?! |
| SP6: Yes, and we are already doing it. |
| SP4: We are already doing it, Jenny! We need to prioritize. |
| SP6: I can tell you that the next available time for treatment of these patients is Dec 4th! (. . .) |
| SP5: But we are there where we have to do it. |
| All are silent. |
| SP11 And with our spirits thus uplifted (ironic). |

In this sequence, doctors' frustration was observed. They challenged the decision to follow the will of patients who–at the cost of other possibly more needy patients–would not accept a delay in their treatment. The potential internal collegial conflict was witnessed in phrases such as "No offense, but. . ." and "Not to go on about it, but. . ." which were speech acts signaling potential frustration while keeping a civil tone. SP4's use of phrases such as–"we have been asked to", "we have been told to", and "we are not allowed to" indicated that the decision was made further up in the hierarchy, e.g., the hospital management. In this way, she positioned herself and the head of department as not responsible and thus not to be blamed for implementing this decision. On the contrary, she positioned them as fighting for the department's interest against the hospital management's unreasonable decisions, thereby marking a "them and us". The final ironic speech act "and with our spirit thus uplifted. . ." could have the function of establishing a sense of community by expressing shared frustration, and taking the potential internal conflict out of the episode: irony positioned all in the same unfortunate boat of being required to act in ways they found questionable.

Another recurrent theme related to *the flow in and out of the department*, *e.g.*, people arriving to or departing from the department. Students, registrars and specialists came into and left the department, which led to the acts of welcoming and bidding farewell, like "I want to introduce Lynn who is our new secretary and started May 1st. It's good to have you here". In one situation, a consultant, also a professor, prepared his colleagues for the arrival of new students in the department later that day:

We are about to welcome just over twenty new students who will enter this room when we leave. I had the pleasure of evaluating with the previous group and they actually gave really good feedback to the department. First, they said: 'Wow, it was so much better than we thought it would be.' (Everyone laughs). And then they said that there is a really good and welcoming atmosphere and that they are allowed to do a lot and they feel welcome. And it really is important that they leave us with that impression. They also say that the organization we have created with teams and that we group them into small groups, dyads, work really well. They would have liked to have more of the same —and I guess one always does. That's that little trick—that when you have a student by your side, try to involve them even more. It makes their day if they are just allowed to stand and hold some stick in the OR or if they get some task to solve. It doesn't take much before they think it has been a really good day. So, it's the small tricks (. . .) So take good care of them. (S1)

Explicating that "we" are about to welcome students marked the induction of new medical students. As seen, the consultant articulated the potential challenge of involving the medical students enough. Positioning the students as being happy just to feel involved as opposed to actually being involved could be seen as devaluing them in their absence. However, the purpose underlying this seemed to be that of enticing his fellow busy senior colleagues to involve the students to a greater extent, with the expression "small tricks".

All departments spent time addressing people coming in and leaving. This could be seen to have dual functions. It had the practical purpose of collaboration, as it is easier to contact the secretary if you have already met her. It had, moreover, an additional and almost ritualistic function of establishing or maintaining an identity and position as a new member of the "we" of the department.

## Patient handover

Patient handover was a central part of all morning reports where different types of positioning were observed. In an episode, a departing on-call doctor (JD14), who was in residency, began his patient handover by presenting a patient who had been admitted during the night:

JD14 During a procedure xx, one lung collapsed. They deflated her on both sides, and she arrived here with a chest compression system. During the episode, she suffered cardiac arrest and was resuscitated. However, she developed subcutaneous emphysema and literally hyper-inflated like a balloon (shows with his hands). She is completely hyper-inflated. We started cardiopulmonary support. Then, she started bleeding from the airways and from the drainage systems, nose and the oral cavity. (. . . .)
She is completely hyper-inflated, and it is extremely difficult to support the airways and circulation–we tried to do xx but did not succeed. Right now, she is stable–but still totally hyper-inflated.
Anyway, I promised to present the case here at the conference in order to find out where to go now. Since she is bleeding in her lungs an (n or an ZZ would be nice–however, she is totally hyper-inflated so it might not be possible.

SP11 It looks quite tricky. . .

JD14 Yes, it is really difficult.

JD14 But I must say that we [the team on call] were six hours to begin with and we are happy to be where we are now. It has been. . ..

A scan is shown on the screen in the conference room

JD14 Look, doesn't she have a monstrous xx?

JD12 Well, it is a little twisted here.

Different positioning dynamics were observed here. The on-call doctor used "we" throughout the story, and "asking advice" was formulated as "I promised to present the case here", presumably as opposed to framing it as an individual challenge of not knowing what to do. By doing this, he positioned himself as a colleague equal to the rest: at the request from someone

outside, the "we" in the conference room are asked to find a solution, "What offers do *we* have?" In addition, he framed the story in a dramatic way, repeating, "she hyper-inflated" five times, each time accompanied by a dramatic hand gesture. This functioned as an implicit emotional appeal and request for collegial recognition of the challenging situation he found himself in. This became even more explicit when he stated: "But I must say that even if we [the team on call] were six hours behind to begin with, we are happy to be where we are now". As seen, his colleagues' responses to his handover focused on speech acts such as giving advice and asking additional biomedical questions. Only consultants spoke, thereby acting as senior staff giving advice, which could be seen as positioning themselves by following the storyline of members of a hierarchy. None of them addressed the emotional appeal, however alert and willing they were to answer medical questions, as witnessed in the change of subject following the departing on-call doctors' explicit appeal which was answered by "but look [at the scan], doesn't she have an unusually large xxx?" In this case, both storylines were at play simultaneously and created challenges, because the rights and obligations in the different positions clashed. As a "good colleague" one is obliged to express support for colleagues who have been challenged in their work. By contrast, senior doctors are obliged to give advice. The emotional core of the departing on-call doctor's patient handover where he positioned himself as an equal colleague was left hanging and did not receive recognition by the senior doctors. They followed the storyline of preserving hierarchy.

| |
|---|
| JD21: The patient presented with high blood pressure, nausea, vomiting, headache, and dizziness, so I started treatment XX. |
| A consultant and specialist doctor question the treatment.<br>SP22: Couldn't you have started a bit more gently with for instance 2.5 mg rather than 5 mg? |
| JD21: She got 5 mg yesterday and 10 today. But it is, after all, prescribed by xx (the specialist registrar). |
| SP23, SP24 and SP25 all at the same time: Yes yes, well well. |
| JD21: But with these symptoms. . . well I do find it difficult with this treatment! |
| SP24: Yes, well I think the first thing is to find out if there is an indication for acute treatment. |
| JD21: Yes. |
| SP22: Yes (nods). |
| SP24: And what is that. . . that is when the patient is hypertensive in relation to acute pulmonary edema, [mentioning of other symptoms, inaudible], headache with cerebral affection. . . and perhaps that isn't the situation here, or . . . uhh . . ..then perhaps. . . |
| JD21: But she had those symptoms. . .? |
| SP24: Yes. |
| JD21: She is actually vomiting. |
| SP24: Yes, and if we can relate that to the hypertension; but can we? So, if there isn't an acute treatment needed, you could choose a more hesitant strategy—see what happens. . . Well, patients who are admitted with headache and dizziness are not rare in our department, but which came first, the chicken or the egg? So, is it because they are scared?. . . You put them to bed and see what happens. |
| JD21: I just don't think I understand what an acute need is, then, if it is not nausea, vomiting, headache, and dizziness! (with emphasis) |
| SP24: Well, it isn't . . . uhh . . . |
| JD21: Well, what is it then? (with emphasis) |
| SP24: Well, there has to be affected consciousness or neurological deficits. |
| JD21: Well, okay. |
| SP24: It is a mild form of hypertension, it isn't severe symptoms. |
| JD21: No. |
| The discussion about hypertensive headache continues. |
| JD21: It is just not easy to determine when you are standing there with the symptoms! |
| The others nod and confirm. |

In another handover situation, different positioning dynamics were seen. The departing on-call doctor (JD21) was a first-year registrar. She presented a patient admitted the night before:

In this interaction, the departing on-call doctor positioned herself as a competent member of the hierarchy: she enumerated the key symptoms, explained the treatment, and explained how she had consulted the relevant specialist registrar before starting the treatment. When faced with criticism from senior staff members, she reacted strongly. She implicitly attacked them with the statement, "I just don't think I understand what an acute indication is, then, if it is not nausea, vomiting, headache, and dizziness! (with emphasis)" implying that they had not taught her correctly, as they were obliged to do. She thus challenged their right to criticize her. The conflict developed gradually, with her interrupting their explanations, until they defused their initial criticism by accepting her statement "It is just not easy to determine when you are standing there with the symptoms!" The situation changed from a battle about who had not lived up to their duties, to the challenging situation of finding out whether a patient was in acute need of treatment.

## Humor

Humor was another repeated element in the morning report. There were improvised elements that were significant for the interaction. The dominant form of humor was the collegial joke, that is, jokes about colleagues inside the department (occasionally about colleagues from other specialties, and, more rarely, about patients). Furthermore, despite all having the right to tell a joke in theory, in reality junior doctors abstained from telling jokes, but merely laughed at jokes from more senior doctors.

The following is an example of how laughing about a colleague or one's own doubts about treatment and examinations could contribute to the creation of a sense of community.

---

An on-call registrar tells about his nightshift. (JD24)

And in OR, we had a couple of patients. The first is Mrs. Jones. She looked like someone bleeding, right when I got here yesterday. So, we had a look inside, but actually there was no sign of bleeding, so that was fine. Then later they called another patient, Mrs. Brown, and then (an on-call registrar laughs and smiles), of course, I was a bit more reluctant (all laughs), but the symptoms continued, so we took a look inside and she was bleeding. . .

---

The colleagues' laughter might illustrate identification with the doctor and the trickiness of the situation he was in, which only insiders would recognize. The general laughter implicitly illustrated collegial support, and thus could be interpreted as contributing to a sense of collegial community.

## The working plan

An element in all departments' morning reports was going through the work plan (roster) of the day. It varied who conducted this (consultant responsible for education, on-call registrar, etc.), but all names in each function were mentioned. This had the obvious function of ensuring that all staffing functions were covered. In addition, appointing doctors in the room made the colleagues in the room "visible", which, considering the variation in the flow of staff, made it possible to know who was who. Especially newcomers had the chance to correct their name as in "No, Catherine sits over there, I'm Sandra". Thus, it had a kind of ritualistic function of being a named and visible member of the "we" of the department.

## Closing and the minutes after

How the "closing" of the morning report happened varied. Occasionally, it was marked explicitly, e.g., by the person conducting the report saying, "Enjoy your work, and for those who are going home: sleep well". However, the dominant form was an implicit recognition of the morning report coming to an end, making everybody leave the conference room. The minutes after the "closing" were characterized by the same informal interactions as in the prelude; however, it was more work-oriented, and involved asking things like:"You were to join me today, right? Let's meet in 15 minutes" or "Did you have time to see my email?" As such, this "after phase" made it possible to plan supervisions, work assistance, etc., informally.

## Discussion

This study shows the social interactions and communication during the morning report and how the elements in a morning report may contribute to the formation of professional identity and the creation of a sense of community. By positioning oneself and others as a collegial "we" and equal members of a department/specialty, as well as "having a place in a hierarchical community", two different storylines were observed, and thus two modes of community were enacted in the morning report.

Using positioning theory, we found that the morning report involved repeated interactional patterns. Other studies have also identified a predictable pattern of events in the morning report [5,7]. We add to their findings by showing that the stringency in performing the different elements had the function of making participation easy, especially for newcomers (or latecomers). One gets to know the rhythm of the morning report quickly as it is simple and repeated in its structure, and easily learned as in a simple version of a "line dance". In their study of surgery teams, Satava and Hunter used choreography to capture the implicit, discursive and bodily patterns in operation teams [35]. Congruent with this, we found that none of the departments explicated the pattern, e.g., by writing it down, yet the doctors still performed it. Speaking metaphorically, they danced the dance without explicit knowledge of the choreography. It would be interesting to explore doctors' perspectives on how "dancing" the morning report affected them becoming a member of a specialty and thus their professional identity. However, that is beyond the scope of this study. Further research that included interviews would be needed to capture this aspect.

Hilligoss et al. have provided a micro-level analysis of the complexity and situatedness involved in handovers. They exclude other activities relating to the morning report [18], and thus we add to their study by identifying the socializing effects of these occasions of handover, acknowledging that handover as an element in the morning report is an expression of the cultural norms and values in the department that are negotiated and maintained over time [36]. In a study of the quality of handovers, Rayo et al observed that while interactive questioning is necessary to ensure quality, communication occasionally takes the form of critique [37]. We add to their findings by showing that junior doctors may perform such critique by questioning whether their seniors live up to their educational duties, which may in turn create conflict.

Our identification of the storyline of the collegial "we" and equal members of a department/specialty storyline confirms findings from a study showing that residents value the camaraderie dimension of the morning report [8]. We add to this by providing a micro-level analysis of how the social interactions that are perceived as camaraderie unfold.

We share the aim of researchers who have used social identity theory to shed light on "in-group" and "out-group" interactions in medical education in order to understand the ways in which individual behavior is influenced by group memberships [16,38,39]. In support of our findings, Burford argues that members of a group have positive attitudes towards in-group

members and negative towards out-group members [16]. Sharing his theoretical approach, Hewett et al. concluded that "specialty" is the most salient professional identity for hospital doctors: doctors refer to themselves and their fellow specialists collectively as members of their specialty group and categorize specialists from other departments as "out-group" members [38]. Our identification of the storyline of being an equal "in-group" member of the community confirms this. However, employing positioning theory allowed us to show that the morning report is both a space for learning to be a member of a collective "we" and *simultaneously* a place where the individual doctor learns his or her place in a hierarchical community, and most important that these storylines are different in nature and sometimes clash. The lens of positioning theory thus captures the tensions and dynamics of the duality of the in-group position, as such.

The focus of this study has not been on how the morning report contributes to workplace learning. However, our findings concerning social interaction and positioning resonate with learning theories, such as Lave and Wenger's understanding of community of practice [40]. They see learning as a social activity that takes place in the interaction between expert and novice, between trainer and trainee [40]. During such processes, the trainee doctor becomes a specialist by learning both the formal competencies and curriculum embedded in the specific community so that they acquire and behave according to the (tacit) "cultural dimensions" of the profession [17,41]. Our findings shed new light on this social dimension and how specific tacit cultural dimensions are enacted and performed, i.e., how different types of "we" are established. Some studies have investigated the format and content of morning reports [12] and others how to optimize the formal learning outcomes through making the more "silent" participants (i.e., junior doctors) more active [42], but further research of both formal and informal learning patterns in the morning report is needed.

It has been observed that the discourse of the morning report involves an ideology where faculty physicians deflect the discourse of patients presented by residents that deviates from medical ideology through critical humor or sarcasm, thus silencing alternative worldviews [7]. How patients are presented in morning reports has not been our focus. However, we observed other community-making functions of humor and irony among colleagues, which may supplement their findings. Other studies have pointed out that humor and laughter can enable the integration of individuals into a group, which supports our findings [43]. However, our study has not captured the more excluding and aggressive functions of laughter in delineating in-group and out-group positions [43].

Theoretically, the use of concepts from speech act theory could be questioned, as it has been criticized for imposing artificial categories on speech [44]. However, we argue that because of its focus on intentional actions, positioning theory makes it possible to capture the dynamics of social episodes such as the morning report.

Our study has limitations. Studies have shown that registrars often criticize the tone, leadership, and learning environment of morning reports [1]. Our study does not elucidate this dimension, as we have only observed the performed actions, and not asked participating doctors about their perspective. In addition, interviews would have shed light on the intentions behind and perceived effects of the interactions which was beyond the scope of this observation-based study. Methodologically, the use of video observations raises the question about participant reactivity, i.e. if the method caused changes in participants' behaviors [45]. However, in our continued dialogue with participants during the data collection, we did not hear any reports about the morning report being different than usual. A limitation is that our study took place in a limited number of departments in Denmark at only two different hospitals. This prevented us from achieving full data saturation. We found repeated interactions and variations, but we cannot rule out the possibility that new themes would have appeared had we

conducted more observations. Furthermore, Denmark has a "flatter" hierarchy than other countries and thus the results might not be generalizable. Still, our comprehensive data material ensured rich data, and the use of video observation where all authors participated in the analysis enabled us to revisit the recordings several times in order to validate the analysis. In addition, we provide a theoretical development by using the framework of positioning theory to gain new insights into the collegial interactions among hospital doctors. This was strengthened by the interdisciplinary group of researchers from the humanities and medicine, allowing for multiple perspectives that enriched the analysis, which is a recommended research strategy [7].

## Conclusion

The morning report is a part of community building, despite it not being explicated as such. It unfolds as a "dance" of repeated elements in a complex collegial space. Within this complexity, the morning report is a space for positioning oneself and others as a collegial "we" and equal members of a department and specialty, at the same time as "having a place" in a hierarchal community. These two storylines and modes of positioning showed that the creation of community was dynamic, as shifts occurred between them.

## Acknowledgments

We wish to thank the hospital departments and all members of staff who opened their doors to us and participated in the process.

We wish to thank student assistants Frederik Søelberg and Astrid Hoppe for their contributions.

## Author Contributions

**Conceptualization:** Jane Ege Møller, Mads Skipper, Lone Sunde, Anita Sørensen, Thomas Balslev, Pernille Andreassen, Bente Malling.

**Data curation:** Jane Ege Møller.

**Formal analysis:** Jane Ege Møller, Mads Skipper, Lone Sunde, Anita Sørensen, Thomas Balslev, Pernille Andreassen, Bente Malling.

**Funding acquisition:** Jane Ege Møller, Mads Skipper, Lone Sunde, Anita Sørensen, Thomas Balslev, Pernille Andreassen, Bente Malling.

**Investigation:** Jane Ege Møller, Pernille Andreassen, Bente Malling.

**Methodology:** Jane Ege Møller, Mads Skipper, Lone Sunde, Anita Sørensen, Thomas Balslev, Pernille Andreassen, Bente Malling.

**Project administration:** Jane Ege Møller.

**Validation:** Jane Ege Møller, Mads Skipper, Lone Sunde, Anita Sørensen, Thomas Balslev, Pernille Andreassen, Bente Malling.

**Writing – original draft:** Jane Ege Møller, Bente Malling.

**Writing – review & editing:** Jane Ege Møller, Mads Skipper, Lone Sunde, Anita Sørensen, Thomas Balslev, Pernille Andreassen, Bente Malling.

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
