## [Decision Letter · Decision Letter 0]

26 Sep 2022

PONE-D-22-07184The choreography of morning reports: forming community and socializing into a clinical department.PLOS ONE

Dear Dr. Møller,

Thank you for submitting your manuscript to PLOS ONE. After careful consideration, we feel that it has merit but does not fully meet PLOS ONE’s publication criteria as it currently stands. Therefore, we invite you to submit a revised version of the manuscript that addresses the points raised during the review process.

You are most fortunate to have prompted insights from two highly respected researchers, whose different theoretical and methodological perspectives have nevertheless resulted in divergent recommendations.  I have also carefully read your manuscript.  

We all share the goal to promote the dissemination of high-quality qualitative research. In cases of divergent recommendations, my standard practice is to select the implicit median---major revisions that MUST address all reviewer points. My general advice consistent with the empirical and methodological emphasis of PLOS ONE, is to 1)  be much more cautious in the leap from observations to potential explanation (e.g., humor) 2) showcase the data, taking the pressure off of extensive theoretical interpretation, and 3) focus on surprising/insightful/practically relevant findings that go beyond well-established social identity theory or common-sense folk psychology.  Although novelty is not a requirement for publication, it is in your best interest to capture the reader's attention.  Below I identify some of the key areas for improvement, but I also refer you to the reviewers’ more specific comments. 

Consistent with Reviewer 1, I find the organization scheme to be incoherent.  Keep it simple and map the research questions onto results headers, with sub-headers as needed.   You claims would also gain credibility from explicit alignment with the analytic template.Positioning theory as portrayed in Figure 1 (position, storyline, action type), is only loosely connected to the data and analysis.  While these elements and their interactions could be made more explicit with subheaders in the results, PLOS one is not a theoretically oriented outlet.   A grounded theory perspective could be more fitting, albeit data sparsity threatens claims of saturation, and I am not aware of any resources for triangulation.   My strong suspicion is that there is a resource management theme in your data--- see lines 284 and 346.Consistent with Suchman, resist the temptation to attribute purposeful coordination in behavior, particularly for these data, where you identify multiple “patterns” across departments. “Choreography” is much too intent-laden for the observations you have reported---else where is the analysis that shows the absence of certain patterns from an un-choreographed process?   And wherever you use a quantifier, you must present the methods and quantitative analysis that supports these assertions including the procedure for the resolution of coder disagreement.The reference to ‘speech acts’ does not seem consistent with the Austin/Searle notion, so worth defining if you retain this language. Regarding conversation analysis, I point you to the less-well known but personal favorite, David Gibson.  Gibson, D. R. (2008). How the outside gets in: Modeling conversational permeation. *Annu. Rev. Sociol*, *34*, 359-384.  Some of reviewer 1’s questions are identified there. Review and refer to the extensive literature that Reviewer 2 suggested.  This will help you highlight new findings.  Your insight that the physical environment may constrain the exchange is consistent with the reviewer’s general concern for other constraints on behavior, such as technology.

We look forward to receiving your revised manuscript.

Kind regards,

Valerie L. Shalin, Ph.D

Academic Editor

PLOS ONE

Journal Requirements:

Reviewers' comments:

Reviewer's Responses to Questions

**Comments to the Author**

1. Is the manuscript technically sound, and do the data support the conclusions?

Reviewer #1: No

Reviewer #2: Yes

2. Has the statistical analysis been performed appropriately and rigorously? 

Reviewer #1: N/A

Reviewer #2: Yes

3. Have the authors made all data underlying the findings in their manuscript fully available?

Reviewer #1: No

Reviewer #2: No

4. Is the manuscript presented in an intelligible fashion and written in standard English?

Reviewer #1: Yes

Reviewer #2: Yes

5. Review Comments to the Author

Reviewer #1: The article is based on video recordings of morning reports. This is certainly a strength, as it offers the researchers to be at the same time very close to what is happening and allowing repeated views, and thus close study. However, the results do not show the harvest of such a close study. As the methodology of the study is only briefly explained and important information is missing, it is not clear why this is the case. Hence, although containing several worthwhile findings, in my view, there are too many limitations to warrant international publication.

The study is designed to explore “1) how social interactions and communication are displayed in morning reports, and 2) how this influences the construction of professional identity and socialization into the clinical department’s community.” The first objective is only partly achieved, the second hardly at all. With only observation data and only 9 hours of observation (my estimation), it is not at all an easy task. The result section is structured through subheadings which do not relate to the research question, are different in nature and do not have an obvious logic. This does not make it easier to see what answers to the research question the authors are giving. Important concepts are not explained. It is unclear why the actors use the metaphor of choreography. I could not detect anything choreographic about the behavior of the participants. In the study the authors use as inspiration for this concept, there was a clear bodily component. I do not see coordinated and expressive movement, to me the core characteristics of a choreography. The meaning and use of position theory is not clearly explained. The authors use “pattern” frequently, but it is unclear what their definition of pattern is, as it is used in cases were the normal (dictionary) meaning to me does not seem to apply. I have the impression that it is used too easily.

The description of the aim of the study (examining how they influence the construction of professional identity and socialization into the community of the clinical department.) suggests the study of a process. The results do not reveal such a process.

The second research question (how this influences the construction of professional identity and socialization into the clinical department’s community) to me, seems to be answered by conjectures, based on other sources than on interpretation of the data.

The method of analysis is not clear. The acts and actors are mentioned, but the (methodological) framework is unclear. Moreover, in the result section “findings” appear that I cannot relate to the (type of) data that were available (i.e., observation data only). How can a researcher from the video recordings of morning reports conclude that “informal collegial chat and small talk contributed to creating or maintaining interpersonal relations”? Or :“By laughing, the colleagues recognized that they appeared to have been positioning him as one of the outsiders (maybe, annoying administrators) who introduce such exercises. Laughing at his joke thus symbolized him being a member of the group of equal colleagues.” Or: “Despite this uneven distribution, the speech act of telling jokes contributed to an informal atmosphere and sense of community.” I do not see what data are used for such statements. The last phase of the analysis is a “theoretical analysis using the framework of positioning theory”. It is not clear what this entails, nor why this framework was chosen and how it contributes to answering the research questions.

Important information regarding the methods of the study is missing. We know how large the staff of the departments was, but not how many persons (of which grades) participated. We do not know how the morning reports were selected, who had to decide whether they could be videotaped and how the participants were informed about the aim of the recording and how the researcher introduced her/himself (which can, of course, influence the behavior of the participants). It is further unclear how much data were collected.

The meaning of storyline in this context is not clear. The storylines are labelled “alternative”, and at the same time it is said that they occur together. This, to me, seems to be contradictory.

The findings lack depth. They are limited to a superficial description of what happens. It is f.i. described by what means the meeting is started (in light of the research questions rather trivial information) but not who decides that it is time to start, and whose activities or presence are taken into account or disregarded.

A synthesis, a real analysis, seems to be absent. The report of the findings, sometimes, creates an impression of being anecdotal or to even miss the point (as in the case of Mrs. Brown’s doctor where laughing is qualified as joke). The observation that “(t)he degree to which humor was used varied slightly between the morning reports, depending on who was present. “is not taken further. Present as user of humor or as an authority who can impose -implicitly or explicitly - what is acceptable, or who defines the “climate”? Of course, as no interviews were held, it is very difficult to decide - but then it should not have been mentioned.

In the analysis, there is very little attention to differences and what they are related to. This contributes to the impression of an anecdotal nature of findings.

Reviewer #2: This article describes findings from an exploratory study analyzing videos of morning report. The exploratory qualitative analysis was based in a theoretical framing of positioning theory, which emphasizes social interactions and communication. Concepts grounding the analysis included choreography, speech acts, social episodes, and cultural norms, narrative conventions, and storylines. Four different hospitals participated. A key finding was that each hospital was unique in the ordering of repeated elements that were choreographed similar to a dance. Two alternative storylines were identified across the four hospitals, one emphasizing equality and one emphasizing hierarchical positioning. Implications are that morning reports contribute to professional identity and socialization into the medical community.

Overall, this is a rigorously conducted study with a strong theoretical foundation that provides interesting insights into how morning rounds vary across hospitals and the storylines that underlie the rituals across hospitals, and how those storylines contribute to professional identity and socialization in a hierarchical medical community with strong similarities across the world.

In the spirit of using peer review to improve scientific manuscripts, suggested revisions, primarily to relate these interesting findings to related articles not yet cited, are:

1. In addition to the literature cited, morning rounds have been researched with respect to information technology support. An aspect that might be of interest is how the nursing flow sheet documents decisions for discharge planning that involve the patient and are driven by physicians, but generally physicians or patients do not have access to the software. One wonders if the placement in the hierarchy and stovepiping of software is due in large part to the professional identity – and if so, what implications this might have for designing software such as having logins and look-and-feel that maintain identity while granting access appropriately? There might be similar insights related to information technology from this article:

a. Gurses, A. P., & Xiao, Y. (2006). A systematic review of the literature on multidisciplinary rounds to design information technology. Journal of the American Medical Informatics Association, 13(3), 267-276.

2. The theoretical framework and concepts used in this study seem to me to insightfully extend the ones considered in this systematic review of ICU teamwork. It might be helpful to state this, and/or identify what is similar about the frameworks and terminology in this article:

a. Dietz, A. S., Pronovost, P. J., Mendez-Tellez, P. A., Wyskiel, R., Marsteller, J. A., Thompson, D. A., & Rosen, M. A. (2014). A systematic review of teamwork in the intensive care unit: what do we know about teamwork, team tasks, and improvement strategies?. Journal of critical care, 29(6), 908-914.

3. It appears that only physicians were included in the morning rounds in this study. My understanding is that typically nurses, pharmacists and social workers are often included by design in morning rounds in the Intensive Care Unit. Perhaps this distinction could be noted in the introduction where comments are made about internal medicine and how morning reports from North America. Although there are probably many references for this, one conference paper that explains the functions of having nurses listen in on physician rounds at four units is:

a. Patterson, E. S., Hofer, T., Brungs, S., Saint, S., & Render, M. L. (2006, October). Structured interdisciplinary communication strategies in four ICUs: An observational study. In Proceedings of the Human Factors and Ergonomics Society Annual Meeting (Vol. 50, No. 10, pp. 929-933). Sage CA: Los Angeles, CA: SAGE Publications.

4. I find the insights about laughter fascinating. Leora Horwitz did an analysis of the use of laughter in ‘training’ medical trainees about how dysfunctional the healthcare system in the United States is during verbal handover communications. (Unfortunately, I could not find the paper quickly.) My understanding is that speech acts do not include laughter? I wonder if perhaps there is a classic reference for in-group, out-group biases relating to laughter? A quick google search turned up this, which I am not sure is the best set of references: “Laughter can promote the integration of new individuals into an already‐present group structure (Gamble 2001), but can also play a role in delineating in‐group and out‐group boundaries (laughter’s “dark side;” Panksepp 2000) by establishing “exclusionary group identities” and by being directly aggressive towards members of the out‐group (Eibl‐Eibesfeldt 1989; Pinker 1997).” This is from: Gervais, M., & Wilson, D. S. (2005). The evolution and functions of laughter and humor: A synthetic approach. The Quarterly review of biology, 80(4), 395-430.

5. The integration of patient handover in morning report might be a unique feature as compared to the United States, where sign-outs are distinguished from interdisciplinary (multidisciplinary) rounds.

6. The reaction to criticism on line 356 is an interesting one, about challenging whether educational duties had been lived up to. This category was not found in this observational analysis of verbal handovers by physicians, nurse practitioners, and nurses in the ICU, although some of the other elements are shown in this example (which you might consider citing as a similar observation):

a. Rayo, M. F., Mount-Campbell, A. F., O'Brien, J. M., White, S. E., Butz, A., Evans, K., & Patterson, E. S. (2014). Interactive questioning in critical care during handovers: a transcript analysis of communication behaviours by physicians, nurses and nurse practitioners. BMJ quality & safety, 23(6), 483-489.

7. Some aspects of the community-building function of the handover relate to and extending Framing 7 in this article

a. “Framing 7, cultural norms, relates to how group values (instantiated as social norms for acceptable behavior) in an organization or suborganization are negotiated and maintained over time”

b. Patterson, E. S., & Wears, R. L. (2010). Patient handoffs: standardized and reliable measurement tools remain elusive. The joint commission journal on quality and patient safety, 36(2), 52-61.

Thank you for providing me the opportunity to review this submission. I look forward to seeing a revision.

6. PLOS authors have the option to publish the peer review history of their article (what does this mean?). If published, this will include your full peer review and any attached files.

Reviewer #1: **Yes: **Maria Grypdonck

Reviewer #2: No

---

## [Author Response · Author response to Decision Letter 0]

29 Nov 2022

Dear Editors of Plos One

We thank you for the opportunity to submit a revised version of our article. We find the feedback has enabled us to improve the quality of the article.

Below we have addressed the comments and suggestions from Editor and reviewers point by point. We hope that our revision is sufficient, if not please let us know if you need any other alterations.

Best regards,

Jane Ege Møller

 Comment Our answer

 Editor 

1 Ed: My general advice consistent with the empirical and methodological emphasis of PLOS ONE, is to 1) be much more cautious in the leap from observations to potential explanation (e.g., humor) Thank you for this very useful advice. We have carefully scrutinized the Results section for opinions and prejudice. Please see the more complete answer to the reviewer’s comment in comment box 11.

2 Ed: 2) showcase the data, taking the pressure off of extensive theoretical interpretation, Thank you for this comment. We have carefully read the Results section and changed the theoretical interpretations, e.g., by modifying or clarifying them throughout the section. Furthermore, we have indicated the level of analysis that we aim for with reference to Steiner Kvale (see answer to comment 18).

3 Ed: focus on surprising/insightful/practically relevant findings that go beyond well-established social identity theory or common-sense folk psychology. Although novelty is not a requirement for publication, it is in your best interest to capture the reader's attention. Thank you for bringing this to our attention: We have deleted selected ‘describing’ parts of the Results sections and clarified other parts, thereby hoping to have sharpened the analysis. 

We have difficulty in seeing how the analysis can be understood as not going beyond social identity theory as we have pointed out in the Discussion section how exactly the positioning theory framework enables us to do this. In order to revise this suggestion in more detail we would need more information from the editor/reviewers.

4 Ed: Consistent with Reviewer 1, I find the organization scheme to be incoherent. Keep it simple and map the research questions onto results headers, with sub-headers as needed. You claims would also gain credibility from explicit alignment with the analytic template.

 We agree that the scheme can appear to be incoherent and have modified it. In the Methods section, we have elaborated on the analysis and synthesis including an explanation on the seeming discrepancy between the analysis template and the headings in the Results section. 

5 Ed: Positioning theory as portrayed in Figure 1 (position, storyline, action type), is only loosely connected to the data and analysis. While these elements and their interactions could be made more explicit with subheaders in the results, PLOS one is not a theoretically oriented outlet. A grounded theory perspective could be more fitting, albeit data sparsity threatens claims of saturation, and I am not aware of any resources for triangulation. My strong suspicion is that there is a resource management theme in your data--- see lines 284 and 346.

 Thank you for making this point. While we do not agree that our analysis is only loosely connected to the framework of positioning theory, we recognize that this has not been visible in our Results section. In the revised version, we have added the three concepts of storyline, positioning and (speech) act throughout the Results section in order to make the basis of the analysis clear.

We appreciate grounded theory; however, we find that using this approach would involve a different study, and therefore have not changed the foundation of our analysis.

6 Ed: Consistent with Suchman, resist the temptation to attribute purposeful coordination in behavior, particularly for these data, where you identify multiple “patterns” across departments. “Choreography” is much too intent-laden for the observations you have reported---else where is the analysis that shows the absence of certain patterns from an un-choreographed process? And wherever you use a quantifier, you must present the methods and quantitative analysis that supports these assertions including the procedure for the resolution of coder disagreement.

 We have changed the use of pattern. Please see answer 14 for a more elaborated answer

We agree that the concept of choreography is intent-laden and therefore have deleted it from the title and methods section.

We have carefully reread the Results section and in the cases where we found the use of quantification, we have changed this. If we have overlooked other quantifiers, we would be happy to change these, but we would need more information.

7 Ed: The reference to ‘speech acts’ does not seem consistent with the Austin/Searle notion, so worth defining if you retain this language. 

 Thank you for this point: we have elaborated the meaning of how positioning theory draws on the notion of speech acts in the Methods section and in the Discussion section.

8 Ed: Regarding conversation analysis, I point you to the less-well known but personal favorite, David Gibson. Gibson, D. R. (2008). How the outside gets in: Modeling conversational permeation. Annu. Rev. Sociol, 34, 359-384. Some of reviewer 1’s questions are identified there. 

 Thank you for mentioning this reference. We find that including it in our study makes it possible for us to discuss the speech act concept in positioning theory in a more nuanced way. We hope that this has made it more visible how positioning theory is in accordance with Gibson’s critique in the following quote:

Speech act sequence analysis has fallen on hard times. Such analyses tend to deliver results that are either obvious or uninterpretable, in the latter case leading us to become suspicious of the coding rules involved. (…) Some other way must thus be found to categorize content while avoiding the pitfalls of classic speech act analysis, particularly the imposition of artificial categories and the assumption that speech acts are neatly enclosed within individual utterances or speaking turns

9 EdReview and refer to the extensive literature that Reviewer 2 suggested. This will help you highlight new findings. Your insight that the physical environment may constrain the exchange is consistent with the reviewer’s general concern for other constraints on behavior, such as technology.

 We thank Reviewer 2 for the extensive list of references and find that most of them add to our study and make the discussion more nuanced, and they have thus been included. In the few cases where we have left a reference out, we have explained the reason for this.

 Reviewer #1: 

10 The article is based on video recordings of morning reports. This is certainly a strength, as it offers the researchers to be at the same time very close to what is happening and allowing repeated views, and thus close study. However, the results do not show the harvest of such a close study. As the methodology of the study is only briefly explained and important information is missing, it is not clear why this is the case. Hence, although containing several worthwhile findings, in my view, there are too many limitations to warrant international publication. Thank you for this comment. We agree that video recordings is a strength, both because of the mentioned closeness to the material and the opportunity for repeated views. In addition, this method is quite rare in studies of morning reports, which in our view is another strength.

11 The study is designed to explore “1) how social interactions and communication are displayed in morning reports, and 2) how this influences the construction of professional identity and socialization into the clinical department’s community.” The first objective is only partly achieved, the second hardly at all. With only observation data and only 9 hours of observation (my estimation), it is not at all an easy task. We appreciate the opportunity to reconsider the research questions as suggested:

The study design is explorative, using video observations of 43 morning conferences in 4 different departments – in all 15,5 hours of recordings (see answer to comment 20 and revised table 1). In the analysis the way participants speak and behave just before, under and just after the morning conference is described using the lenses of positioning theory. Thus, the first research question has been answered through the results. 

We acknowledge that observations are not enough to display or describe the influences behavior and speech during morning conferences have on professional identity formation and socialization into the medical community. Therefore, the research question has been reformulated to: 1) how social interactions and communication are displayed in morning reports, and 2) what that means for the formation of professional identity and socialization into the clinical department’s community.

In addition, the formulation of results has been thoroughly revised in order not to be too opinionated or prejudiced (also see comment number 18) and at the same time keeping the understanding and interpretation from the analysis and synthesis:

Example: “By laughing, the colleagues recognized that they appeared to have been positioning him as one of the outsiders (maybe, annoying administrators) who introduce such exercises. Laughing at his joke thus symbolized him being a member of the group of equal colleagues.”

Has been reformulated to:

“By laughing, the colleagues probably recognized that they appeared to have been positioning him as one of the outsiders (e.g. administrators) who introduce such exercises. Laughing at his joke thus may function as recognition of him as being a member of the group of equal colleagues”

We hope that these reformulations throughout the Results section make it clearer that the research questions are answered by the results.

12 The result section is structured through subheadings which do not relate to the research question, are different in nature and do not have an obvious logic. This does not make it easier to see what answers to the research question the authors are giving. In the Methods section, we have elaborated on the development and use of the analysis template and further described the analysis and synthesis including an explanation of how the elements in the analysis template turned into the headings in the Results section.

Furthermore, the answers of the research questions have been described in the Discussion section. 

We do not find that it will add clarity to structure the Results sections according to the research questions, and so this has not been changed. We find that this view adheres with the qualitative research tradition.

13 Important concepts are not explained. It is unclear why the actors use the metaphor of choreography. I could not detect anything choreographic about the behavior of the participants. In the study the authors use as inspiration for this concept, there was a clear bodily component. I do not see coordinated and expressive movement, to me the core characteristics of a choreography We appreciate the opportunity to revise this part of the study. We have reconsidered the way in which we use choreography and agree that this could imply a clearer bodily focus. We have therefore removed this from the title and deleted it in the Methods sections.

14 The meaning and use of position theory is not clearly explained. The authors use “pattern” frequently, but it is unclear what their definition of pattern is, as it is used in cases were the normal (dictionary) meaning to me does not seem to apply. I have the impression that it is used too easily Thank you for pointing out that we have used the word “pattern” in more than one way – namely to describe practice, design, order, sequence, characteristics as well as more figuratively. In order to make the meaning more explicit, we have scrutinized the article and changed the word to more explicit wordings – and only kept the word “pattern” when it refers to “characteristics”. 

In addition, we have elaborated on the meaning and use of positioning theory in the Methods section.

15 The description of the aim of the study (examining how they influence the construction of professional identity and socialization into the community of the clinical department.) suggests the study of a process. The results do not reveal such a process. We appreciate this comment.

As described under Comment 11, we are fully aware that observation cannot reveal a process. Therefore, we have changed the research question – please see under Comment 11.

16 The second research question (how this influences the construction of professional identity and socialization into the clinical department’s community) to me, seems to be answered by conjectures, based on other sources than on interpretation of the data. Thank you:

Research question 2 has been revised. Furthermore, a thorough revision of the results as suggested under comment 18 has been performed.

It has thus been made clear where the results are based on interpretation through the analysis and synthesis of data.

In the Discussion section, we have made it clearer how we answer the questions.

17 The method of analysis is not clear. The acts and actors are mentioned, but the (methodological) framework is unclear. Thank you for pointing this out. We have added information in the Methods section to clarify the methodological steps.

18 Moreover, in the result section “findings” appear that I cannot relate to the (type of) data that were available (i.e., observation data only). How can a researcher from the video recordings of morning reports conclude that “informal collegial chat and small talk contributed to creating or maintaining interpersonal relations”? Or :“By laughing, the colleagues recognized that they appeared to have been positioning him as one of the outsiders (maybe, annoying administrators) who introduce such exercises. Laughing at his joke thus symbolized him being a member of the group of equal colleagues.” Or: “Despite this uneven distribution, the speech act of telling jokes contributed to an informal atmosphere and sense of community.” I do not see what data are used for such statements. Thank you for pointing to this issue. We have read these sections and modified as suggested. As we provide a theoretically informed analysis of the observed interactions, positions, (speech) acts and storylines, we find our level of interpretation valid. In Kvale’s often quoted work, he makes the distinction between the three contexts or levels of interpretation: (a) the interviewee’s own understanding of what he is saying, (b) a more general, commonsense conception of the meaning of what he is saying, and (c) a theoretical level of interpretation of what he is saying.

Phenomenology + Pedagogy Volume 6 Number 2 1988. 

19 The last phase of the analysis is a “theoretical analysis using the framework of positioning theory”. It is not clear what this entails, nor why this framework was chosen and how it contributes to answering the research questions.

 We have added information about this step, unfolded the description of positioning theory and highlighted why this framework is suitable for answering the research questions.

20 Important information regarding the methods of the study is missing. We know how large the staff of the departments was, but not how many persons (of which grades) participated. We do not know how the morning reports were selected, who had to decide whether they could be videotaped and how the participants were informed about the aim of the recording and how the researcher introduced her/himself (which can, of course, influence the behavior of the participants). It is further unclear how much data were collected.

 Thank you for the opportunity to make this clear.

The Methods section has been thoroughly revised to provide data regarding the selection of departments and morning reports. Each morning the “photographer” introduced him/herself and the project until all participants were familiar with the project. 

Table 1 has been revised and now shows how many hours of video recordings were performed, and the average number of participants in the morning reports has been added.

The information regarding consent is provided in the Ethics section.

21 The meaning of storyline in this context is not clear. The storylines are labelled “alternative”, and at the same time it is said that they occur together. This, to me, seems to be contradictory Thank you for pointing this out. While there is no theoretical contradiction between having alternative storylines appearing together, we realize that we have not adequately described why this (a strength of the framework, in our view) is not a contradiction. We have tried to clarify this in the Methods section.

22 The findings lack depth. They are limited to a superficial description of what happens. It is f.i. described by what means the meeting is started (in light of the research questions rather trivial information) but not who decides that it is time to start, and whose activities or presence are taken into account or disregarded. We have deleted selected descriptions of the Results sections and clarified other parts thereby hoping to have sharpened the analysis. We have difficulty in seeing how the analysis can be understood to not go beyond social identity theory as we have pointed out in the Discussion section how exactly the positioning theory framework enables us to do this. To revise this suggestion in more detail we would need more information from the editor/reviewers.

23 A synthesis, a real analysis, seems to be absent. The report of the findings, sometimes, creates an impression of being anecdotal or to even miss the point (as in the case of Mrs. Brown’s doctor where laughing is qualified as joke). 

The observation that “(t)he degree to which humor was used varied slightly between the morning reports, depending on who was present. “is not taken further. 

Present as user of humor or as an authority who can impose -implicitly or explicitly – what is acceptable, or who defines the “climate”? Of course, as no interviews were held, it is very difficult to decide – but then it should not have been mentioned.

 Thank you for this comment.

With regards to the issue of lacking synthesis and real analysis please see our answer to Comment 12.

To meet the relevant criticism from the reviewer, the section about humor has been revised. We have deleted the sentences about variations. Likewise, the citation referred to (Mrs. Brown…) has been elaborated to make the point of the example stand out more clearly.

24 In the analysis, there is very little attention to differences and what they are related to. This contributes to the impression of an anecdotal nature of findings. Our focus has not been a comparative analysis of the differences between the departments (instead, we are interested in variation) therefore this aspect does not receive attention.

With regards to the issue of the anecdotal nature of findings, we hope that our elaboration of the Methods sections has made it clear that the findings are not anecdotal but the result of careful analysis.

 Reviewer #2 

25 Reviewer #2: This article describes findings from an exploratory study analyzing videos of morning report. The exploratory qualitative analysis was based in a theoretical framing of positioning theory, which emphasizes social interactions and communication. Concepts grounding the analysis included choreography, speech acts, social episodes, and cultural norms, narrative conventions, and storylines. Four different hospitals participated. A key finding was that each hospital was unique in the ordering of repeated elements that were choreographed similar to a dance. Two alternative storylines were identified across the four hospitals, one emphasizing equality and one emphasizing hierarchical positioning. Implications are that morning reports contribute to professional identity and socialization into the medical community. Thank you for this summary.

26 Overall, this is a rigorously conducted study with a strong theoretical foundation that provides interesting insights into how morning rounds vary across hospitals and the storylines that underlie the rituals across hospitals, and how those storylines contribute to professional identity and socialization in a hierarchical medical community with strong similarities across the world.

 Thank you for the positive feedback.

27 1. In addition to the literature cited, morning rounds have been researched with respect to information technology support. An aspect that might be of interest is how the nursing flow sheet documents decisions for discharge planning that involve the patient and are driven by physicians, but generally physicians or patients do not have access to the software. One wonders if the placement in the hierarchy and stovepiping of software is due in large part to the professional identity – and if so, what implications this might have for designing software such as having logins and look-and-feel that maintain identity while granting access appropriately? There might be similar insights related to information technology from this article:

a. Gurses, A. P., & Xiao, Y. (2006). A systematic review of the literature on multidisciplinary rounds to design information technology. Journal of the American Medical Informatics Association, 13(3), 267-276. This is a very interesting point. We did not focus on the use of technology as described in the study, and we did not observe any technological support that influenced the interaction. In addition, we did not study multidisciplinary settings. Therefore, we find it difficult to relate this comment to our findings and have therefore not included it.

28 2. The theoretical framework and concepts used in this study seem to me to insightfully extend the ones considered in this systematic review of ICU teamwork. It might be helpful to state this, and/or identify what is similar about the frameworks and terminology in this article:

a. Dietz, A. S., Pronovost, P. J., Mendez-Tellez, P. A., Wyskiel, R., Marsteller, J. A., Thompson, D. A., & Rosen, M. A. (2014). A systematic review of teamwork in the intensive care unit: what do we know about teamwork, team tasks, and improvement strategies?. Journal of critical care, 29(6), 908-914 We thank the reviewer for mentioning this issue. We have read the Dietz review and while it is a very interesting article, we find that its specific focus on ICU and teamwork is very different from our study, as is the framework and terminology. If we were to include this in the article, we would need some elaboration from the reviewer.

29 3. It appears that only physicians were included in the morning rounds in this study. My understanding is that typically nurses, pharmacists and social workers are often included by design in morning rounds in the Intensive Care Unit. Perhaps this distinction could be noted in the introduction where comments are made about internal medicine and how morning reports from North America. Although there are probably many references for this, one conference paper that explains the functions of having nurses listen in on physician rounds at four units is:

a. Patterson, E. S., Hofer, T., Brungs, S., Saint, S., & Render, M. L. (2006, October). Structured interdisciplinary communication strategies in four ICUs: An observational study. In Proceedings of the Human Factors and Ergonomics Society Annual Meeting (Vol. 50, No. 10, pp. 929-933). Sage CA: Los Angeles, CA: SAGE Publications.

 Correct – only physicians participate in the morning report in the Danish context. It is now further stressed in the “Setting” section that only physicians participate in the morning reports under investigation.

We acknowledge the benefits of doing multidisciplinary conferences, and most other conferences in Denmark are multidisciplinary.

The article underlines these benefits and was interesting reading.

30 4. I find the insights about laughter fascinating. Leora Horwitz did an analysis of the use of laughter in ‘training’ medical trainees about how dysfunctional the healthcare system in the United States is during verbal handover communications. (Unfortunately, I could not find the paper quickly.) My understanding is that speech acts do not include laughter? I wonder if perhaps there is a classic reference for in-group, out-group biases relating to laughter? A quick google search turned up this, which I am not sure is the best set of references: “Laughter can promote the integration of new individuals into an already‐present group structure (Gamble 2001), but can also play a role in delineating in‐group and out‐group boundaries (laughter’s “dark side;” Panksepp 2000) by establishing “exclusionary group identities” and by being directly aggressive towards members of the out‐group (Eibl‐Eibesfeldt 1989; Pinker 1997).” This is from: Gervais, M., & Wilson, D. S. (2005). The evolution and functions of laughter and humor: A synthetic approach. The Quarterly review of biology, 80(4), 395-430.

 Thank you for drawing our attention to this aspect and to the studies.

We have added Gervais’s points in our Discussion section.

32 5. The integration of patient handover in morning report might be a unique feature as compared to the United States, where sign-outs are distinguished from interdisciplinary (multidisciplinary) rounds. Thank you for drawing our attention to the fact that different countries have different cultures and practices. In the section called Setting we have further elaborated on the Danish context in order to make it clearer for an international audience that may have other routines.

33 6. The reaction to criticism on line 356 is an interesting one, about challenging whether educational duties had been lived up to. This category was not found in this observational analysis of verbal handovers by physicians, nurse practitioners, and nurses in the ICU, although some of the other elements are shown in this example (which you might consider citing as a similar observation):

a. Rayo, M. F., Mount-Campbell, A. F., O'Brien, J. M., White, S. E., Butz, A., Evans, K., & Patterson, E. S. (2014). Interactive questioning in critical care during handovers: a transcript analysis of communication behaviours by physicians, nurses and nurse practitioners. BMJ quality & safety, 23(6), 483-489. Thank you for pointing to this study. We agree with your point and have added it in the Discussion section as we find that it strengthens the discussion.

34 7. Some aspects of the community-building function of the handover relate to and extending Framing 7 in this article

a. “Framing 7, cultural norms, relates to how group values (instantiated as social norms for acceptable behavior) in an organization or suborganization are negotiated and maintained over time”

b. Patterson, E. S., & Wears, R. L. (2010). Patient handoffs: standardized and reliable measurement tools remain elusive. The joint commission journal on quality and patient safety, 36(2), 52-61. Thank you for this reference. We have added this study and their point about cultural norms influencing patient handover to the Discussion section.

---

## [Decision Letter · Decision Letter 1]

19 Jan 2023

PONE-D-22-07184R1How doctors build community and socialize into a clinical department through morning reports. A positioning theory studyPLOS ONE

Dear Dr. Møller,

Thank you for submitting your manuscript to PLOS ONE. After careful consideration, we feel that it has merit but does not fully meet PLOS ONE’s publication criteria as it currently stands. Therefore, we invite you to submit a *modestly *revised version of the manuscript that addresses the points raised during the review process.

Congratulations on a much improved submission.  Special congratulations on reversing the skepticism of a highly respected reviewer.  I'm sure I can speak for all reviewers in expressing our appreciation for the attention you gave to our comments.    The absence of a raw data file is understandable when livelihoods and professional reputations are at stake. Consider adding a few more example excerpts if they make sense, and encourage readers to contact you with specific questions on the corpus.  In your final revision, please indicate all of the limitations and wording concerns that Reviewer 1 identifies, such as the absence of convergent evidence.  I agree that you can manage the generalization issue better, though I have a slightly different take on how to address this.   In my opinion, we could use a better description of the idiosyncracies of the environments that you observed.  I feel like we are missing context, which could be provided in your setting description.  I don't understand the general admission procedure and hence the doctor's defensive response to being challenged.  I surely don't understand the availability of hospital beds in Denmark, and how close to the edge of capacity the Danish system functions.  I have no insight into the prevalence of administrative intrusion that motivated the humorous exchange. Yes, this actually emphasizes the specificity of your study, but I find this to be the best way to explain the inevitable differences in an attempt to replicate your work.  Though important revisions, they are not difficult to evaluate.  Please be sure these are identifiable in marked text. It is not my intention to request a third round of external reviews.   

Please submit your revised manuscript by  February, 28, 2023.  If you will need more time than this to complete your revisions, please reply to this message or contact the journal office at plosone@plos.org. Please include the following items when submitting your revised manuscript:A rebuttal letter that responds to each point raised by the academic editor and reviewer(s). You should upload this letter as a separate file labeled 'Response to Reviewers'.A marked-up copy of your manuscript that highlights changes made to the original version. You should upload this as a separate file labeled 'Revised Manuscript with Track Changes'.An unmarked version of your revised paper without tracked changes. You should upload this as a separate file labeled 'Manuscript'.If applicable, we recommend that you deposit your laboratory protocols in protocols.io to enhance the reproducibility of your results. Protocols.io assigns your protocol its own identifier (DOI) so that it can be cited independently in the future. For instructions see: https://journals.plos.org/plosone/s/submission-guidelines#loc-laboratory-protocols. Additionally, PLOS ONE offers an option for publishing peer-reviewed Lab Protocol articles, which describe protocols hosted on protocols.io. Read more information on sharing protocols at https://plos.org/protocols?utm_medium=editorial-email&utm_source=authorletters&utm_campaign=protocols.

We look forward to receiving your revised manuscript.

Kind regards,

Valerie L. Shalin, Ph.D

Academic Editor

PLOS ONE

Journal Requirements:

Reviewers' comments:

Reviewer's Responses to Questions

**Comments to the Author**

1. If the authors have adequately addressed your comments raised in a previous round of review and you feel that this manuscript is now acceptable for publication, you may indicate that here to bypass the “Comments to the Author” section, enter your conflict of interest statement in the “Confidential to Editor” section, and submit your "Accept" recommendation.

Reviewer #1: (No Response)

Reviewer #2: All comments have been addressed

2. Is the manuscript technically sound, and do the data support the conclusions?

Reviewer #1: Partly

Reviewer #2: Yes

3. Has the statistical analysis been performed appropriately and rigorously? 

Reviewer #1: N/A

Reviewer #2: Yes

4. Have the authors made all data underlying the findings in their manuscript fully available?

Reviewer #1: No

Reviewer #2: Yes

5. Is the manuscript presented in an intelligible fashion and written in standard English?

Reviewer #1: Yes

Reviewer #2: Yes

6. Review Comments to the Author

Reviewer #1: Many of the recommendations have been followed up, and where they are not, usually a satisfactory explanation is given, indicating that, at least, there are (good) reasons for what is done. Recommendations made by other reviewers to re-analyze are not followed up, and I can understand that.

The methods are described in more detail and easier to follow. By this, the findings “speak” more, and gain in interest.

However:

In my view, the limitations of the study are not pointed out well enough. A major limitation is that recordings without interviews do give any information about intentions nor about effects. In your previous version, it was clear that the temptation is great to make that kind of conclusions. And even in this version, there are some left. You say that, according to Kvale, interpretations can be theory based. However, in social science such theoretical interpretations do not establish facts, and what is said should remain recognizable as interpretation (f. i. by stating “this can be seen as; or “this can be expected to lead to…”, and not using an indicative (they do, the feel…) or “this establishes” or “this leads to”. This is something that still needs correction in a few places.

Generalization is an issue. You say that your findings are not generalizable since you examined only two hospitals. To begin with you pointed out that the morning reports in Denmark have a specific form. Therefore, transferability to other countries is low (assuming that in all Danish hospitals the format, intentions and practice are similar). Generalizability cannot be obtained as in no way saturation is achieved (and theory development minimal). In my view, it would be more correct to say that it is uncertain whether the findings portray what is the case in other settings as it is unclear whether the existing variation has been captured.

Reviewer #2: I thank the authors for their responsiveness to all comments. I find it particularly interesting that only physicians participate in rounds in Denmark, as this varies unit by unit within a hospital, and certainly across hospitals. It even varies by month depending on what the attending physician wishes to do.

7. PLOS authors have the option to publish the peer review history of their article (what does this mean?). If published, this will include your full peer review and any attached files.

Reviewer #1: **Yes: **maria grypdonck

Reviewer #2: No

---

## [Author Response · Author response to Decision Letter 1]

15 Mar 2023

Comment Reviewer/Editor Answer

Ed: The absence of a raw data file is understandable when livelihoods and professional reputations are at stake. Consider adding a few more example excerpts if they make sense, and encourage readers to contact you with specific questions on the corpus. We agree that a raw data file would have been ideal, and we thank for your understanding of why it is not possible We find it difficult to add example excerpts at this stage, however we agree to encourage readers to contact us. We have tried to add these in information we add in the editorial system.

Ed: In your final revision, please indicate all of the limitations and wording concerns that Reviewer 1 identifies, such as the absence of convergent evidence. We have revised accordingly. Please se details below.

Ed: I agree that you can manage the generalization issue better, though I have a slightly different take on how to address this. In my opinion, we could use a better description of the idiosyncracies of the environments that you observed. I feel like we are missing context, which could be provided in your setting description. I don't understand the general admission procedure and hence the doctor's defensive response to being challenged. I surely don't understand the availability of hospital beds in Denmark, and how close to the edge of capacity the Danish system functions. I have no insight into the prevalence of administrative intrusion that motivated the humorous exchange. Thank you for this suggestion. We have tried to clarify the context by adding the following: 

To better understand our study, it is relevant to know that the healthcare system in Denmark is primarily public, and general practitioners are gatekeepers for secondary and tertiary care. As a citizen, you need a referral from your general practitioner to be admitted to a hospital. The hospitals have limited capacity, and there are generally waiting lists. However, by law, various “packages of care” have been introduced. Thus, a patient with symptoms of serious or acute illness (for example cancer) can bypass the waiting list in order to be treated fast. Likewise, there are rules ensuring that as a patient, your symptoms will be considered within 30 days from referral. The hospitals are financed by one of five regions and therefore, the regional authorities prefer to keep treatment inside the region, since treatment in another region would entail extra payment. There can be an inbuilt conflict between doctors, who want the best treatment for their patients, and the management, who wants the most treatment for the finances they have at disposal from the government..

Rev 1: In my view, the limitations of the study are not pointed out well enough. A major limitation is that recordings without interviews do give any information about intentions nor about effects. In your previous version, it was clear that the temptation is great to make that kind of conclusions. And even in this version, there are some left. You say that, according to Kvale, interpretations can be theory based. However, in social science such theoretical interpretations do not establish facts, and what is said should remain recognizable as interpretation (f. i. by stating “this can be seen as; or “this can be expected to lead to…”, and not using an indicative (they do, the feel…) or “this establishes” or “this leads to”. This is something that still needs correction in a few places.

 We have carefully read through the manuscript and changed the (two) places were this could be misinterpreted. If we are acquired to revise this point further we would need specific guidance as to where in the text.

Rev 1: Generalization is an issue. You say that your findings are not generalizable since you examined only two hospitals. To begin with you pointed out that the morning reports in Denmark have a specific form. Therefore, transferability to other countries is low (assuming that in all Danish hospitals the format, intentions and practice are similar). Generalizability cannot be obtained as in no way saturation is achieved (and theory development minimal). In my view, it would be more correct to say that it is uncertain whether the findings portray what is the case in other settings as it is unclear whether the existing variation has been captured. We have tried to addressed these points in the discussion in the section that now reads: 

Our study has limitations. Studies have shown that registrars often criticize the tone, leadership, and learning environment of morning reports [1]. Our study does not elucidate this dimension, as we have only observed the performed actions, and not asked participating doctors about their perspective. In addition, interviews would have shed light on the intentions behind and perceived effects of the interactions which was beyond the scope of this observation-based study. Methodologically, the use of video observations raises the question about participant reactivity, i.e. if the method caused changes in participants’ behaviors [45]. However, in our continued dialogue with participants during the data collection, we did not hear any reports about the morning report being different than usual. A limitation is that our study took place in a limited number of departments in Denmark at only two different hospitals. This prevented us from achieving full data saturation. We found repeated interactions and variations, but we cannot rule out the possibility that new themes would have appeared had we conducted more observations. Furthermore, Denmark has a “flatter” hierarchy than other countries and thus the results might not be generalizable. Still, our comprehensive data material ensured rich data, and the use of video observation where all authors participated in the analysis enabled us to revisit the recordings several times in order to validate the analysis. In addition, we provide a theoretical development by using the framework of positioning theory to gain new insights into the collegial interactions among hospital doctors. This was strengthened by the interdisciplinary group of researchers from the humanities and medicine, allowing for multiple perspectives that enriched the analysis, which is a recommended research strategy [7]

---

## [Editor Report · Decision Letter 2]

13 Apr 2023

How doctors build community and socialize into a clinical department through morning reports. A positioning theory study

PONE-D-22-07184R2

Dear Dr. Møller,

We’re pleased to inform you that your manuscript has been judged scientifically suitable for publication and will be formally accepted for publication once it meets all outstanding technical requirements. 

Kind regards,

Valerie L. Shalin, Ph.D

Academic Editor

PLOS ONE

Additional Editor Comments (optional):

PLEASE ADDRESS THE FOLLOWING IN YOUR COPY EDITED SUBMISSION:  1) CLARIFY YOUR WILLINGNESS TO DISCUSS THE DATA WITH READERS IN THE DATA AVAILABILITY SECTION. 2) INDICATE PSEUDONYMS FOR ANY NAMES THAT PERSIST IN THE EXAMPLES.   
---

## [Editor Report · Acceptance letter]

27 Apr 2023

PONE-D-22-07184R2 

How doctors build community and socialize into a clinical department through morning reports. A positioning theory study 

Dear Dr. Møller:

I'm pleased to inform you that your manuscript has been deemed suitable for publication in PLOS ONE. Congratulations! Your manuscript is now with our production department. 

Kind regards, 

on behalf of

Dr. Valerie L. Shalin 

Academic Editor

PLOS ONE